# Improved artificial potential-based formation control of Multi-USV systems for collision and obstacle avoidance under GPS attacks

1st Sen Cheng
*School of Navigation*
*Wuhan University of Technology*
Wuhan, China
sencheng@whut.edu.cn

2nd Yue Yang
*School of Navigation*
*Wuhan University of Technology*
Wuhan, China
yueyang@ieee.org

3rd Xiaochen Li*
*Tianjin Research Institute for Water*
*Transport Engineering, M.O.T.*
Tianjin, China
Lixch@tiwte.ac.cn

*Abstract*—This paper proposes a formation control method for multi-USV systems with collision and obstacle avoidance under GPS attacks. First, the external environmental disturbance and nonlinear dynamics of the multi-USV systems are handled using fuzzy logic systems. Second, a protocol for the formation control of multi-USV systems under GPS attacks is designed. The collision and obstacle avoidance issues are addressed by an improved artificial potential field (APF) method. Finally, simulation examples demonstrate the effectiveness of the proposed control method.

*Index Terms*—Formation control, unmanned surface vessels, artificial potential field methods, GPS attacks

## I. INTRODUCTION

With the rapid advancement of artificial intelligence and computer technology, unmanned surface vessel (USV) systems have emerged as critical assets for enhancing operational efficiency and mitigating personnel risks [1], [2], [3], [4]. As a burgeoning marine technology, multi-USV systems are increasingly showcasing their potential in diverse fields including marine science, resource exploration and military applications. The formation of multi-USV systems can accomplish tasks more efficiently through coordinated efforts compared to a single USV. Additionally, formation control benefits from enhanced flexibility and robustness [5], [6], [7]. Therefore, the formation control of multi-USV systems have garnered significant attention among scientific disciplines [8]. With the widespread application of multi-USV systems, various formation control methods have been developed, such as behavior-based [9], virtual structure [10], artificial potential field [11], [12], basic graph theory [13], [14], [15], the leader-follower [16], among others. Among these, the leader-follower method receives considerable interest due to its simplicity and scalability [17].

In recent years, scholars have shown significant interest in research related to multi-USV systems formation control.

This paper is supported by the National Key Research and Development Program of China (2023YFB2603800), the National Natural Science Foundation of China (NSFC) under Grant No. 52031009, and the Fundamental Research Funds for the Central Universities (WUT: 2024IVA046).(Corresponding author: Xiaochen Li).

However, when maneuvering in complex environment, collision may occur among multi-USVs. Many advanced control algorithms have been investigated, including neural networks [18], [19], disturbance observer [18], dynamic surface control [18], artificial potential field [11], [12] , minimal learning parameter approaches [19], and back-stepping [20]. The APF is an effective method for collision avoidance [21], [22], besides some problems have been prompted researchers to propose improvements [23]. According to the potential function described in [21], [22], an adaptive formation control method for a class of multi-agent systems was designed in [24]. Currently, maritime vessels rely extensively on GPS signals provided by commercial systems. However, the civilian GPS signals expose multi-USVs to vulnerabilities, particularly from GPS spoofing attacks. These attacks transmitting fake GPS signals, which can deceive multi-USV systems and compromise formation control. Such compromises may lead to significant navigational deviation or even collision [25], [26], [27]. Yet, research simultaneously addressing both problems are relatively scarce. With few studies in the literature discussing the impact of GPS spoofing attacks on the formation control, collision and obstacle avoidance mechanisms of multi-USV systems.

Motivated by the previous discussion, the research on multi-USV systems exists many challenges. This paper addresses the formation control problem for multi-USV systems under GPS attacks with input saturation and model uncertainties. The effectness of the proposed method is verified by thoery and demonstrated by simulation examples.

The remainder of this paper is structured as follows: Section II presents preliminary and problem formulation, including basic graph theory, the fuzzy logic system and GPS attacks. Section III presents controller design and stability analysis to validate the stability of multi-USV systems. In Section IV , a simulation example is given to verify the effectiveness of the proposed control strategy. Finally, the relevant conclusion of this paper are presented in Section V .

## II. PRELIMINARIES AND PROBLEM FORMULATION

### A. System description

Dynamic modeling of multi-USV systems including three degrees of freedom: surge, sway and yaw. Consider multi-USV systems consisting of one leader and three followers. The mathematical model of ship motion for the $i$th USV is expressed by:

$$\begin{cases} \dot{\eta}_i(t) = J(\eta_i(t)) \nu_i(t) \\ M\dot{\nu}_i(t) + C(\nu_i(t))\nu_i(t) + D(\nu_i(t))\nu_i(t) = \tau_i(t), \end{cases} \quad (1)$$

where

$\eta_i(t) = [\eta_{ix}(t), \eta_{iy}(t), \eta_{iz}(t)]$ denotes as the position vector and head vector of the ith USV,

$\nu_i(t) = [v_{ix}(t), v_{iy}(t), v_{iz}(t)]$ denotes as the velocity vector of the ith USV in the three directions of surge, sway and yaw,

$\tau_i(t) = [\tau_{ix}(t), \tau_{iy}(t), \tau_{iz}(t)]$ denotes as the control input,

$J(\eta_i(t))$ is the rotation matrix used to transform the coordinates, $J^T(\eta_i(t)) = J^{-1}(\eta_i(t))$, which can be represented as:

$$J(\eta_i(t)) = \begin{bmatrix} \cos(\eta_{iz}(t)) & -\sin(\eta_{iz}(t)) & 0 \\ \sin(\eta_{iz}(t)) & \cos(\eta_{iz}(t)) & 0 \\ 0 & 0 & 1 \end{bmatrix}.$$

$M$ denotes as the inertia mass matrix, considering that $\{X_{(\cdot)}, Y_{(\cdot)}, N_{(\cdot)}\}$ are the hydrodynamic parameters.

$$M = \begin{bmatrix} m_{11} & 0 & 0 \\ 0 & m_{22} & m_{23} \\ 0 & m_{32} & m_{33} \end{bmatrix},$$

where $m_{11} = m_{usv} - X_{\dot{x}}, m_{22} = m_{usv} - Y_{\dot{y}}, m_{33} = I_z - N_{\dot{z}}, m_{23} = m_{usv}x_g - Y_{\dot{z}}, m_{32} = m_{usv}x_g - N_{\dot{y}}, Y_{\dot{z}} = N_{\dot{y}}.$

$m_{usv}$ is the mass of USVs,

$x_g$ represents the gravity center of USVs,

$I_z$ is the rotational inertia moment.

$C(\nu_i(t))$ denotes as the Koch force matrix;

$$C(\nu_i(t)) = \begin{bmatrix} 0 & 0 & c_{13} \\ 0 & 0 & c_{23} \\ -c_{13} & -c_{23} & 0 \end{bmatrix},$$

where $c_{13} = -m_{22}v_{iy}(t) - 1/2 \ (m_{23} + m_{32})v_{iz}(t)$, $c_{23} = m_{11}v_{ix}(t)$.

$D(\nu_i(t))$ denotes as the hydrodynamic damping matrix, specific definitions are given below:

$$D(\nu_i(t)) = \begin{bmatrix} d_{11} & 0 & 0 \\ 0 & d_{22} & d_{23} \\ 0 & d_{32} & d_{33} \end{bmatrix},$$

where

$$d_{11} = -X_x - X_{|x|x}|v_{ix}(t)| - X_{|x|xx}v_{ix}^2(t),$$
$$d_{22} = -Y_y - Y_{|y|y}|v_{iy}(t)| - Y_{|z|y}|v_{iz}(t)|,$$
$$d_{23} = -Y_z - Y_{|y|z}|v_{iy}(t)| - Y_{|z|z}|v_{iz}(t)|,$$
$$d_{32} = -N_y - N_{|y|y}|v_{iy}(t)| - N_{|z|y}|v_{iz}(t)|,$$
$$d_{33} = -N_z - N_{|y|z}|v_{iy}(t)| - N_{|z|z}|v_{iz}(t)|.$$

Let

$$C_g = -J(\eta_i(t)) M^{-1} C\left(J^{-1}(\eta_i(t)) v_i(t)\right) J^{-1}(\eta_i(t)),$$
$$D_g = -J(\eta_i(t)) M^{-1} D\left(J^{-1}(\eta_i(t)) v_i(t)\right) J^{-1}(\eta_i(t))$$
$$+ \dot{J}(\eta_i(t)) J^{-1}(\eta_i(t)),$$
$$J(\eta_i(t)) v_i(t) = v_i(t), \eta_i(t) = x_i(t).$$

Then the system (1) can be rewritten as:

$$\dot{x}_i(t) = v_i(t)$$
$$\dot{v}_i(t) = J(x_i(t)) M^{-1}\tau_i(t) + C_g(x_i(t), v_i(t)) v_i(t) \quad (2)$$
$$+ D_g(x_i(t), v_i(t)) v_i(t).$$

Define

$$f_{i0}(x_i(t), v_i(t)) = C_g v_i(t) + D_g v_i(t),$$
$$g_{i0}(x_i(t), v_i(t)) = J(x_i(t)) M^{-1}, \tau_i(t) = sat(u_i(t)).$$

Then the equation (2) is given by:

$$\dot{x}_i(t) = v_i(t)$$
$$\dot{v}_i(t) = f_{i0}(x_i(t), v_i(t)) + g_{i0}(x_i(t), v_i(t)) sat(u_i(t)), \quad (3)$$

where $sat(u_i(t))$ denotes as a control constrained by the input saturation function, as defined by:

$$sat(u_i(t)) = \begin{cases} u_i(t) & |u_i(t)| \le u_i^*(t) \\ sign(u_i(t)) u_i^*(t) & |u_i(t)| > u_i^*(t) \end{cases} \quad (4)$$

where $u_i^*(t)$ denotes as saturation threshold, $sat(u_i(t))$ can be expressed through the smooth function $\zeta(u_i(t))$ as:

$$\zeta(u_i(t)) = \begin{cases} \bar{u}_i(t) \tanh\left(\frac{u_i(t)}{\bar{u}_i(t)}\right) & u_i(t) \ge 0 \\ \underline{u}_i(t) \tanh\left(\frac{u_i(t)}{\underline{u}_i(t)}\right) & u_i(t) < 0 \end{cases} \quad (5)$$

where $\bar{u}_i(t)$ and $\underline{u}_i(t)$ are the upper and lower bounds of the control input $u_i(t)$, $\tanh(u_i(t)) = \frac{e^{u_i(t)} - e^{-u_i(t)}}{e^{u_i(t)} + e^{-u_i(t)}}$.

Define the error function as:

$$|\mu(u_i(t))| = |sat(u_i(t)) - \zeta(u_i(t))| \le \bar{\mu}_i(t). \quad (6)$$

According to the above equation, the nonlinear multi-USV systems (3) can obtain that:

$$\dot{x}_i(t) = v_i(t)$$
$$\dot{v}_i(t) = f_{i0}(x_i(t), v_i(t)) + g_{i0}(x_i(t), v_i(t))(\zeta(u_i(t)) \quad (7)$$
$$+ \mu(u_i(t))).$$

Use the median theorem:

$$\zeta(u_i(t)) = \zeta_i(u_i^o(t))$$
$$+ (u_i(t) - u_i^o(t)) \frac{\partial \zeta(u_i(t))}{\partial u_i(t)}\bigg|_{u_i(t) = u_i^\eta(t)}, \quad (8)$$

where $u_i^\eta(t) = \eta u_i(t) + (1 - \eta) u_i^o(t), 0 < \eta < 1$, choose $u_i^o(t) = 0$, then $\zeta(0) = 0$. $\zeta(u_i(t))$ can be represented as:

$$\zeta(u_i(t)) = u_i(t) \frac{\partial \zeta(u_i(t))}{\partial u_i(t)}\bigg|_{u_i(t) = u_i^\eta(t)}. \quad (9)$$

Define

$$g_i\left(x_i(t), v_i(t)\right) = g_{i0}\left(x_i(t), v_i(t)\right) \left. \frac{\partial \zeta\left(u_i(t)\right)}{\partial u_i(t)} \right|_{u_i(t) = u_i^\eta(t)}$$

$$f_i\left(x_i(t), v_i(t)\right) = g_{i0}\left(x_i(t), v_i(t)\right) \mu(u_i) + f_{i0}\left(x_i(t), v_i(t)\right).$$

After modification, the system (7) is represented as follows:

$$\begin{aligned} \dot{x}_i(t) &= v_i(t) \\ \dot{v}_i(t) &= f_i\left(x_i(t), v_i(t)\right) + g_i\left(x_i(t), v_i(t)\right) u_i(t). \end{aligned} \tag{10}$$

The system dynamics of the leader USV is presented below:

$$\begin{aligned} \dot{x}_l(t) &= v_l(t) \\ \dot{v}_l(t) &= f_l\left(x_l, v_l, t\right). \end{aligned} \tag{11}$$

**Assumption 1.** *The nonlinear function $f_i\left(x_i(t), v_i(t)\right)$ and the nonlinear matrix function $g_i\left(x_i(t), v_i(t)\right)$ are bounded.*

**Assumption 2.** *$g_i\left(x_i(t), v_i(t)\right)$ is strictly positive definite and eigenvalues $\|\lambda\left(g_i(\cdot)\right)\| > \chi, i = 1, \ldots, n$ are satisfied.*

**Assumption 3.** *The speed of follower USVs and leader USV are bounded.*

**Definition 1.** *The multi-USV systems can realize the desired formation control objective given that the subsequent conditions are satisfied during the formation process: $\lim\limits_{t \to \infty} \|x_i(t) - x_l(t) - \delta_i(t)\| = 0$, $\lim\limits_{t \to \infty} \|v_i(t) - v_l(t)\| = 0, i = 1, \ldots, n$. In the above relation equation, $\delta_i(t)$ denotes the predetermined relative position between the ith USV and the leader USV.*

According to the leader-follower formation structure method, the position and velocity tracking error variables between the follower USVs and the leader USV are defined as follows, respectively:

$$\begin{aligned} \tilde{x}_i(t) &= x_i(t) - x_l(t) - \delta_i(t) \\ \tilde{v}_i(t) &= v_i(t) - v_l(t), \end{aligned} \tag{12}$$

where $\tilde{x}_i(t)$ and $\tilde{v}_i(t)$ denote the position and velocity tracking error of USVs. The dynamic error of multi-USV systems can be obtained by derivation of Eq(12):

$$\begin{aligned} \dot{\tilde{x}}_i(t) &= \dot{\tilde{v}}_i(t) \\ \dot{\tilde{v}}_i(t) &= f_i\left(x_i(t), v_i(t)\right) + g_i\left(x_i(t), v_i(t)\right) u_i(t) - f_l\left(x_l, v_l, t\right). \\ i &= 1, 2, \ldots, n. \end{aligned} \tag{13}$$

Based on the above equations, the system error equation (13) can be obtained:

$$\begin{aligned} \dot{\tilde{Z}}(t) = &- \left[ \begin{bmatrix} 0_n & -I_n \\ 0_n & 0_n \end{bmatrix} \otimes \mathrm{I}_m \right] \tilde{Z}(t) + \begin{bmatrix} 0nm \\ f_i(t) \end{bmatrix} \\ &+ \begin{bmatrix} 0nm \\ U(t) \end{bmatrix} - \begin{bmatrix} 0nm \\ f_l(x_l, v_l, t) \end{bmatrix}, \end{aligned} \tag{14}$$

where the errors of multi-USV systems are $\dot{\tilde{Z}}(t) = \left[\dot{\tilde{x}}_1^T(t), \ldots \dot{\tilde{x}}_n^T(t), \dot{\tilde{v}}_1^T(t), \ldots, \dot{\tilde{v}}_n^T(t)\right]^T$, the nonlinear vector function are $f_i(t) = \left[f_1^T(t), \ldots, f_n^T(t)\right]^T$, the input of multi-USV systems are $U(t) =$

$\left[\left(g_1(t) u_1(t)\right)^T, \ldots, \left(g_n(t) u_n(t)\right)^T\right]^T$, the input of leader USV are $f_l\left(x_l, v_l, t\right) = \left[f_l^T(t), \ldots, f_l^T(t)\right]^T$, and $\otimes$ denotes Kronecker product.

### B. Basic graph theory

A graph $G = \{\rho, E, W\}$ denotes communication relationships between multi-USVs. Denote all USVs in the formation as the set of non-empty nodes $\rho = \{\rho_1, \rho_2, \ldots, \rho_n\}$, a set of edges $E \subseteq \{(\rho_i, \rho_j) | \rho_i, \rho_j \in \rho\}$, and weighted adjacency matrix $W = [\rho_{ij}] \in \mathrm{R}^{\mathrm{n} \times \mathrm{n}}$. Each element indicates that the nodes can correspond with each other to obtain messages from others. The edge $e_{ij} = (\rho_j, \rho_i)$ denotes the communication of information flow from node $\rho_j$ to node $\rho_i$. Defines the topological neighborhood set $N_i = \{\rho_j \in \rho : w_{ij} \in w, i \neq j\}$. The adjacency matrix $W = [w_{ij}]$ represents the communication relationships between nodes in a graph. The element $w_{ij}$ denotes the edge weight of the corresponding edge $E_{ij}$. Specifically, the diagonal elements $w_{ij} = 0$, and the off-diagonal elements $w_{\mathrm{ij}} > 0$ indicate that there is information flow between nodes $i$ and $j$, otherwise $w_{\mathrm{ij}} = 0$. Furthermore, define the Laplace matrix $L$ of the graph $G$ as follow:

$$L = D - W,$$

where $D = diag\{d_1, d_2, \cdots, d_n\}$ is represented by the diagonal matrix and for the elements of the matrix with $d_i = \sum\limits_{j=1}^{n} w_{ij}, j = 1, 2, \cdots, n$. Let the leader adjacency matrix be defined as $B = diag\{b_1, b_2, \cdots, b_n\}$. Assuming node $i$ and the leader have the connectivity in the graph $G$, $b_i > 0$, otherwise $b_i = 0$.

**Lemma 1.** *Provided that the Laplacian $L = [l_{ij}]$ is irreducible matrix and has dimensions $\mathrm{R}^{n \times n}$, and then the eigenvalues of matrix are positive definite, as presented by:*

$$\hat{L} = L + B = \begin{pmatrix} l_{11} + b_1 & \cdots & l_{1n} \\ \vdots & \ddots & \vdots \\ l_{n1} & \cdots & l_{nn} + b_n \end{pmatrix}$$

*1) GPS system:* GPS consists of 24 satellites, with 21 operational satellites and 3 spares. Ground monitoring comprises three main components: a master control station, injection stations, and monitoring stations. The user segment includes GPS receivers, data processing modules, microprocessors. GPS signals are composed of carrier waves, pseudo-random codes, and navigation message data codes. Each satellite utilizes a reference frequency of 10.23 MHz to generate required $L1/L2$ signals, data codes, and pseudorandom codes. The $L1$ and $L2$ bands in GPS are designated with carrier frequencies of 1575.42 MHz and 1227.60 MHz. GPS is categorized into military and civilian standards, which military signals to $P$-codes and civilian signals to $C/A$ codes.

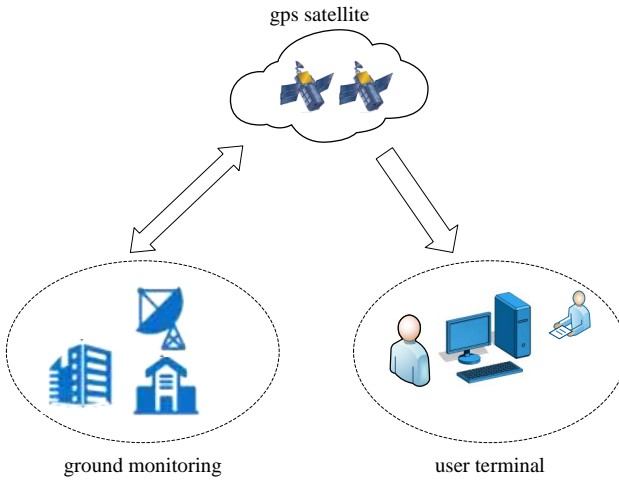

Fig. 1. Components of the GPS system

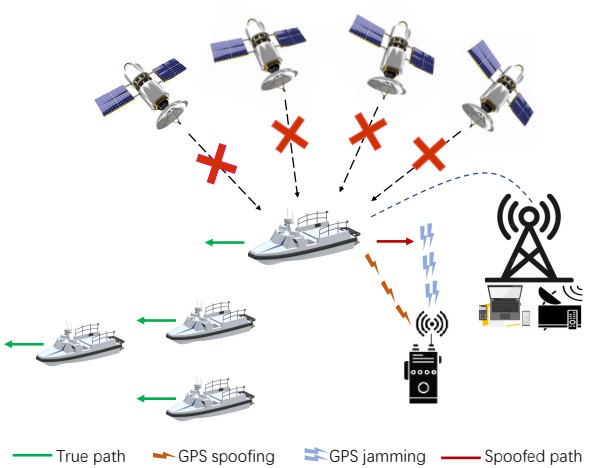

——— True path  ⚡ GPS spoofing  ⚡ GPS jamming  ——— Spoofed path

Fig. 2. How GPS spoofing attacks work

*2) GPS attacks:* GPS attacks include GPS spoofing and jamming attacks. The attacker generates the fake GPS signals so that the target tracks the fake GPS signals, thereby being deceived into an incorrect position. Consider that the communication channel between the leader USV and follower USVs is disrupted. The position of the attacked leader $x_{ls}(t)$ can obtain that:

$$x_{ls}(t) = x_l(t) + Q(t)\gamma, \qquad (15)$$

where $Q(t)$ denotes as continuous or discrete function, $\gamma$ is denoted as a constant displacement bias. In the absence of GPS attacks, the leader position offset $\gamma$ is zero. When the leader USV reaches the desired position, $x_{ls}(t)$ is aligned with the original formation alignment. When GPS attacks is initiated, the $x_{ls}(t)$ vector always differs from the desired position because the vector $\gamma$ always exists. Therefore, the follower USVs are affected by $x_{ls}(t)$ and cannot reach the desired position. Due to the leader formation control, eventually all

follower USVs have a constant deviation from the desired position.

Defines the relative position of the leader USV per moment of change:

$$d_l = \sqrt{(x_l(t+1) - x_l(t))^T (x_l(t+1) - x_l(t))}.$$

The GPS attacks are indicated if at a certain point in time $d_l$ undergoes a large number of changes above a certain threshold $\varphi$, that is $d_l > \varphi$. Due to the target position being known, the leader USV position state is continuously adjusted by means of correction coefficients

$$x_{lr}(t) = x_{ls}(t) + k_r\tilde{\gamma}, \qquad (16)$$

where $x_{lr}(t)$ is the recovered trajectory position, $k_r$ is the correction coefficient, and $\tilde{\gamma}$ is the correction vector.

*C. Fuzzy logic system*

The fuzzy logic system consists of IF-Then rules, singleton fuzzification, centre average defuzzification. It is well-known that the FLS has the approximation propert. The fuzzy logic system can be represented by the following rules:

If: $z_1$ is $A_1^l$, $z_2$ is $A_2^l$, ..., $z_n$ is $A_n^l$,

Then: $\Phi_1$ is $W_1^l$, $\Phi_2$ is $W_2^l$, ..., $\Phi_s$ is $W_s^l$.

where $z = [z_1, z_2, \ldots, z_n]^T \in R^n$ is the input variable, $\Phi = [\Phi_1, \Phi_2, \ldots, \Phi_s]^T \in R^s$ is the output variable, $A_i^l$ in which $i = 1, 2, \ldots, n$, $l = 1, 2, \ldots, N$ denote the $i$th fuzzy set within the $l$th fuzzy rule, $W_j^l$ in which $j = 1, 2, \ldots, s$, $l = 1, 2, \ldots, N$ is the fuzzy set of the output variable $\Phi$ and $N$ is the number of fuzzy rules.

The single instance fuzzifier, the product inference and the center average defuzzifier, $\Phi(z)$ can be expressed as:

$$\Phi(z) = \frac{\sum_{l=1}^{L} \mho^l \prod_{i=1}^{n} \Im_{A_i^l}(z_i)}{\sum_{l=1}^{L} \left[\prod_{i=1}^{n} \Im_{A_i^{(l)}}(z_i)\right]}, \qquad (17)$$

where $L$ deontes the total number of fuzzy rules, $\mho^l$ satisfies $\Im_{W^l}(\mho^l) = \max\{\Im_{W^l}(\Phi(z)) | \Phi(z) \in R\}$.

Let

$$h^l(z) = \frac{\prod_{i=1}^{n} \Im_{A_i^l}(z_i)}{\sum_{l=1}^{N} \left[\prod_{i=1}^{n} \Im_{A_i^l}(z_i)\right]}, \qquad (18)$$

where $\Im_{A_i^l}(z_i)$ is the affiliation function of the fuzzy set $A_i^l$, which is defined as Gaussian function. The total output equation of the fuzzy system is given below:

$$\Phi(z) = R^{*T}H(z) + \varepsilon(z),$$

where $R^*$ is the ideal weight, $H(z)$ is the fuzzy basis function, $\varepsilon(z)$ is the estimation error and threshold is $c$.

**Lemma 2.** *Assuming that $\Phi(z)$ is a continuous function defined on a tight set $\Xi$, there exists the fuzzy logic system satisfying the following inequality:*

$$\sup_{z \in \Xi} \left| \Phi(z) - W^T H(z) \right| \le \ell,$$

where $\ell$ is an arbitrary constant.

## III. CONTROLLER DESIGN AND STABILITY ANALYSIS

### A. Improved artificial potential field

Based on the artificial potential field theory, each follower USVs is assumed to have the same high potential field. The function of the repulsion field designed for collision avoidance is described by:

$$U_{ij} = \begin{cases} \eta \left( \frac{\|\xi_{ij}\|^2 - \rho_{des}^2}{\|\xi_{ij}\|^2 - \rho_{\min}^2} \right)^2 & \rho_{min} < \|\xi_{ij}\| < \rho_{des} \\ 0 & \|\xi_{ij}\| > \rho_{des} \end{cases} \quad (19)$$

where $\eta$ is a positive coefficient, $i$ and $j$ denote the i/jth of the nth USVs, $\|\xi_{ij}\| = \sqrt{(x_i - x_j)^2 + (y_i - y_j)^2}$ for the the Euclidean distance between USVs in the formation, where $[x_i, y_i]^T$ and $[x_j, y_j]^T$ are the positions of the individual USVs, the $\rho_{min}$ and $\rho_{des}$ represent the upper and lower bound through the collision avoidance region.

Define the potential function $U_{ij}$ between the ith USV and the jth USV. The collision avoidance potential field force $F_{ij}$ can be obtained by solving for the negative gradient:

$$F_{ij} = \begin{cases} 4\eta \frac{\xi_{ij}(\rho_{des}^2 - \rho_{\min}^2)(\|\xi_{ij}\|^2 - \rho_{des}^2)}{(\|\xi_{ij}\|^2 - \rho_{\min}^2)^3} & \rho_{\min} \le \|\xi_{ij}\| \le \rho_{des} \\ 0 & \|\xi_{ij}\| > \rho_{des} \end{cases}$$
$$(20)$$

If the actual distance $d_{ij}$ between the ith USV and the jth USV is greater than the desired distance $\rho_{des}$, a gravitational force will be generated to pull the two USVs closer together. However, if $d_{ij}$ is less than $\rho_{des}$, a repulsive force is generated to separate the two USVs. When $d_{ij}$ tends to zero, the resultant force will ensure that the multi-USV systems avoid collision during the interaction motion.

Similarly, in the process of multi-USVs formation control, the existence of obstacles must be taken into account. To mitigate the problem of USVs oscillation, the following potential function for obstacle avoidance is formulated as:

$$U_{rep}(X_p)$$
$$= \begin{cases} \frac{1}{2} K_{\text{rep}} \left( \frac{1}{\rho(X_p, X_{ob})} - \frac{1}{\rho_0} \right)^2 \rho^m(X_p, X_g) & \rho(X_p, X_{ob}) \le \rho_0 \\ 0 & \rho(X_p, X_{ob}) > \rho_0 \end{cases}$$
$$(21)$$

where $m$ is the moderating positive factor. Solving for the negative gradient of the improved repulsive potential field, the repulsive force can be calculated as:

$$F_{rep}(X_p) = -\nabla U_{rep}(X_p)$$
$$= \begin{cases} F_{rep1}(X_p) + F_{rep2}(X_p) & \rho(X_p, X_{ob}) \le \rho_0 \\ 0 & \rho(X_p, X_{ob}) > \rho_0 \end{cases}$$
$$(22)$$

where $F_{rep1}(X_p)$ and $F_{rep2}(X_p)$ can be expressed as

$$F_{rep1}(X_p) = K_{rep} \left( \frac{1}{\rho(X_p, X_{ob})} - \frac{1}{\rho_0} \right) \frac{1}{\rho^2(X_p, X_{ob})}$$
$$\rho^m(X_p, X_g) \frac{\partial \rho(X_p, X_{ob})}{\delta X}$$

$$F_{rep2}(X_p) = -\frac{n}{2} K_{rep} \left( \frac{1}{\rho(X_p, X_{ob})} - \frac{1}{\rho_0} \right)^2$$
$$\rho^{m-1}(X_p, X_{ob}) \frac{\partial \rho(X_p, X_{ob})}{\partial X}.$$

The improved repulsive force vector component $F_{rep1}(X_p)$ is directed from the direction of obstacles away to the USVs. The repulsive force vector component $F_{rep2}(X_p)$ is directed from the USVs to the direction of the target position. When $n \in (0,1)$, $\rho(X_p, X_g) \to 0$, the repulsive component $F_{rep1}(X_p) \to 0$, the repulsive component $F_{rep2}(X_p) \to \infty$. Under the effect of gravitational force $F_{att}(X_p)$ and repulsive force component $F_{rep2}(X_p)$, even if there is an obstacle around the target position.

### B. Controller design

When attacked by GPS fake signals, the position information $x_l(t)$ of the leader USV changes $x_{ls}(t)$, and the attack messages further affect the follower USVs messages. The system dynamics of the leader USV are represented as follows:

$$\begin{aligned} \dot{x}_{ls}(t) &= v_l(t) \\ \dot{v}_{ls}(t) &= f_{ls}(x_{ls}, v_l, t). \end{aligned} \quad (23)$$

Define the position and velocity tracking error variables between the follower and the leader after being attacked by GPS as follows, respectively:

$$\begin{aligned} \tilde{x}_{is}(t) &= x_i(t) - x_{ls}(t) - \delta_i(t) \\ \tilde{v}_{is}(t) &= v_i(t) - v_l(t). \end{aligned} \quad (24)$$

The error system equation (14) can be rewritten:

$$\dot{\tilde{Z}}_s(t) = - \left[ \begin{bmatrix} 0_n & -I_n \\ 0_n & 0_n \end{bmatrix} \otimes I_m \right] \tilde{Z}(t) + \begin{bmatrix} 0_{nm} \\ f_i(t) \end{bmatrix}$$
$$+ \begin{bmatrix} 0_{nm} \\ U_s(t) \end{bmatrix} - \begin{bmatrix} 0_{nm} \\ f_{ls}(x_{ls}, v_l, t) \end{bmatrix}. \quad (25)$$

The definition of all tracking errors in the presence of GPS attacks is given as follows:

$$\begin{aligned} f_{is}^x(t) &= \sum_{j \in N_i} w_{ij} \left( (x_i(t) - x_j(t)) - (\delta_i(t) - \delta_j(t)) \right) \\ &\quad + b_i (x_i(t) - x_{ls}(t) - \delta_i(t)), \\ f_{is}^v(t) &= \sum_{j \in N_i} w_{ij} (v_i(t) - v_j(t)) + b_i (v_i(t) - v_l(t)), \\ i &= 1, 2, \ldots, n. \end{aligned}$$
$$(26)$$

where $w_{ij}$ denotes the elements in row i and column j of the adjacency matrix W of the graph G. $f_{is}^x(t)$ and $f_{is}^v(t)$ are the positional and velocity formation errors under GPS attacks, respectively.

By using the position and velocity tracking error variables between the follower USVs and the leader USV, the above equation (26) is also rewritten as:

$$f_{is}^x(t) = \sum_{j \in N_i} w_{ij}(\tilde{x}_{is}(t) - \tilde{x}_{js}(t)) + b_i \tilde{x}_{is}(t)$$

$$f_{is}^v(t) = \sum_{j \in N_i} w_{ij}(v_i(t) - v_j(t)) + b_i \tilde{v}_i(t). \quad (27)$$

$$i = 1, 2, \ldots, n.$$

The distributed formation controller is designed as follows:

$$u_{is}(t) = \frac{1}{\Gamma}\left(-h_i\left(f_{is}^x(t) + f_{is}^v(t)\right) - \hat{R}_i^T H_i(z)\right), \quad (28)$$

where $\Gamma$ is the positive gain parameter.

$$\dot{\hat{W}} = \varpi_i\left[\left(f_{is}^x(t) + f_{is}^v(t)\right)H_i - \sigma_i \hat{R}_i H_i(z)\right], \quad (29)$$

where $\theta_i$ and $\sigma_i$ are positive parameters.

The controller for multi-USV formation, collision and obstacle avoidance are designed as follows:

$$u_{is}(t) = \frac{1}{\Gamma}\left(-h_i\left(f_{is}^x(t) + f_{is}^v(t)\right) - \hat{R}_i^T H_i(z)\right.$$
$$\left. -\sum_{i=1}^m \nabla_{x_i} U_{rep} - \sum_{j=1, i\neq j}^n \nabla_{x_i} U_{ij}\right). \quad (30)$$

C. Stability analysis

**Theorem 1.** *Consider the system described, and suppose that Assumption 1, Assumption 2, Assumption 3 hold. If the inequality is satisfied by the design parameter $\xi \geq a + b - m_i$, where $a$ , $b$ and $m_i$ are positive parameters, then the control strategy (28) ensures convergence of the follower USVs to their desired positions.*

The Lyapunov candidate function is defined as follows:

$$V(t) = \frac{1}{2}\tilde{Z}_s^T(t)\left(\begin{bmatrix} 2\bar{L}_q & \bar{L}_q \\ \bar{L}_q & \bar{L}_q \end{bmatrix} \otimes \mathrm{I}_m\right)\tilde{Z}_s(t)$$
$$+ \frac{1}{2}\sum_{i=1}^n \varpi_i^{-1}\tilde{R}_i^T(t)\tilde{R}_i(t), \quad (31)$$

where $P = \begin{bmatrix} 2\bar{L}_q & \bar{L}_q \\ \bar{L}_q & \bar{L}_q \end{bmatrix}$, $\bar{L}_q = L_q + B_q$, $N = \begin{bmatrix} 0_n & \mathrm{I}_n \\ 0_n & 0_n \end{bmatrix}$, $\tilde{R}_i = \hat{R} - R^*$ is the error between the ideal and estimated

weights of the the fuzzy logic system.

$$\dot{V}(t) = \frac{1}{2}\dot{\tilde{Z}}_s^T(t)(P \otimes \mathrm{I}_m)\tilde{Z}_s(t) + \frac{1}{2}\tilde{Z}_s^T(t)(P \otimes \mathrm{I}_m)\dot{\tilde{Z}}_s(t)$$
$$+ \sum_{i=1}^n \varpi^{-1}\tilde{R}_i^T(t)\dot{\hat{R}}(t)$$
$$= \frac{1}{2}\tilde{Z}_s^T(t)\left([P^T N + N^T P] \otimes \mathrm{I}_m\right)\tilde{Z}_s(t)$$
$$+ \sum_{i=1}^n \varpi^{-1}\tilde{R}_i^T(t)\dot{\hat{R}}(t)$$
$$+ \tilde{Z}_s(t)[P \otimes \mathrm{I}_m]\left[\begin{matrix} 0_{nm} \\ f_i(t) + U_s(t) - f_{ls}(x_{ls}, v_l, t) \end{matrix}\right]$$
$$= \tilde{Z}_s^T(t)\left(\begin{bmatrix} 0_n & \bar{L}_q \\ \bar{L}_q & \bar{L}_q \end{bmatrix}\right)\tilde{Z}_s(t) + \sum_{i=1}^n \varpi^{-1}\tilde{R}_i^T(t)\dot{\hat{R}}(t)$$
$$+ \sum_{i=1}^n (f_{is}^x(t) + f_{is}^v(t))^T \Phi_i(t)$$
$$- \sum_{i=1}^n (f_{is}^x(t) + f_{is}^v(t))^T f_{ls}(x_{ls}, v_l, t)$$
$$+ \sum_{i=1}^n (f_{is}^x(t) + f_{is}^v(t))^T g_i(x_i(t), v_i(t))u_{is}(t). \quad (32)$$

Approximating nonlinear dynamics of systems with the fuzzy logic system:

$$\dot{V}(t) = \tilde{Z}_s^T(t)\left(\begin{bmatrix} 0_n & \bar{L}_q \\ \bar{L}_q & \bar{L}_q \end{bmatrix}\right)\tilde{Z}_s(t)$$
$$+ \sum_{i=1}^n (f_{is}^x(t) + f_{is}^v(t))^T \left(R_i^{*T}(t)H_i(z_i(t)) + \varepsilon_i(z_i(t))\right)$$
$$- \sum_{i=1}^n (f_{is}^x(t) + f_{is}^v(t))^T f_{ls}(x_{ls}, v_l, t) + \sum_{i=1}^n (f_{is}^x(t) + f_{is}^v(t))^T$$
$$g_i(x_i(t), v_i(t))u_{is}(t) + \sum_{i=1}^n \varpi^{-1}\tilde{R}_i^T(t)\dot{\hat{R}}(t). \quad (33)$$

Introducing the adaptive law $\dot{\hat{R}}_i(t)$ and the controller $u_{is}(t)$, the formula is represented as:

$$\dot{V}(t) = \tilde{Z}^T(t)\left(\begin{bmatrix} 0_n & \bar{L}_q \\ \bar{L}_q & \bar{L}_q \end{bmatrix}\right)\tilde{Z}(t)$$
$$- \sum_{i=1}^n (f_{is}^x(t) + f_{is}^v(t))^T (f_{ls}(x_{ls}, v_l, t) - \varepsilon_i(z_i(t)))$$
$$- \sum_{i=1}^n m_i\|f_{is}^x(t) + f_{is}^v(t)\|^2 - \sum_{i=1}^n \varpi_i\tilde{R}_i^T(t)\hat{R}(t). \quad (34)$$

By assumption and Young's inequality, one gets:

$$-(f_{is}^x(t) + f_{is}^v(t))^T f_{ls}(x_{ls}, v_l, t) \leq a\|f_{is}^x(t) + f_{is}^v(t)\|^2 \frac{\chi}{4a} \quad (35)$$

$$\varepsilon_i(z_i(t))f_{is}^x(t) + f_{is}^v(t)^T \leq b\|f_{is}^x(t) + f_{is}^v(t)\|^2 + \frac{c^2}{4b}, \quad (36)$$

where $a$, $b$, and $c$ are constants, by substitution of equations, one gets:

$$\dot{V}(t) = \tilde{Z}_s^T(t)\left(\begin{bmatrix} 0_n & \bar{L}_q \\ \bar{L}_q & \bar{L}_q \end{bmatrix}\right)\tilde{Z}_s(t)$$
$$+ \sum_{i=1}^{n}(a+b-m_i)\|f_{is}^x(t)+f_{is}^v(t)\|^2 \quad (37)$$
$$+ \sum_{i=1}^{n}\left(\frac{\chi}{4a}+\frac{c^2}{4b}\right) - \sum_{i=1}^{n}\varpi_i\tilde{R}_i^T(t)\hat{R}(t).$$

According to the fact $-jk \le -\frac{1}{2}j^2 + \frac{1}{2}(j-k)^2$, one gets:

$$-\varpi_i\tilde{R}^T(t)\hat{R}_i(t) \le \frac{1}{2}\varpi_i\|R_i^*(t)\|^2 - \frac{1}{2}\varpi_i\left\|\tilde{R}_i(t)\right\|^2. \quad (38)$$

Take $\xi \ge a+b-m_i$, which can be obtained by transforming the above result into the inequality:

$$\dot{V}(t) = \tilde{Z}_s^T(t)\left(\begin{bmatrix} 0_n & \bar{L}_q \\ \bar{L}_q & \bar{L}_q \end{bmatrix}\right)\tilde{Z}_s(t) - \xi\|f_{is}^x(t)+f_{is}^v(t)\|^2$$
$$- \frac{1}{2}\varpi_i\left\|\tilde{R}_i(t)\right\|^2 + \sum_{i=1}^{n}\left(\frac{1}{2}\varpi_i\|R_i^*(t)\|^2 + \frac{\chi}{4a}+\frac{c^2}{4b}\right)$$
$$= -\tilde{Z}_s^T(t)\left(\left(\xi\begin{bmatrix} \bar{L}_q^2 & \bar{L}_q^2 \\ \bar{L}_q^2 & \bar{L}_q^2 \end{bmatrix} - \begin{bmatrix} 0_n & \bar{L}_q \\ \bar{L}_q & \bar{L}_q \end{bmatrix}\right)\otimes I_m\right)\tilde{Z}_s(t)$$
$$- \frac{1}{2}\varpi_i\left\|\tilde{R}_i(t)\right\|^2 + \sum_{i=1}^{n}\left(\frac{1}{2}\varpi_i\|R_i^*(t)\|^2 + \frac{\chi}{4a}+\frac{c^2}{4b}\right) \quad (39)$$

According to the linear matrix inequality $\xi_i\bar{L}_q^2 - (\xi_i\bar{L}_q^2 - \bar{L}_q) = \bar{L}_q > 0$, $\xi_i\bar{L}_q^2 - \bar{L}_q > 0$, one gets:
$$\left[\xi\begin{bmatrix} \bar{L}_q^2 & \bar{L}_q^2 \\ \bar{L}_q^2 & \bar{L}_q^2 \end{bmatrix} - \begin{bmatrix} 0_n & \bar{L}_q \\ \bar{L}_q & \bar{L}_q \end{bmatrix}\right] = \begin{bmatrix} \xi\bar{L}_q^2 & \xi\bar{L}_q^2 - \bar{L}_q \\ \xi\bar{L}_q^2 - \bar{L}_q & \xi\bar{L}_q^2 - \bar{L}_q \end{bmatrix} > 0.$$

Further derivation leads to:

$$\dot{V}(t) \le -\tilde{Z}^T(t)\left([\xi\kappa - \Omega]\otimes I_m\right)\tilde{Z}(t)$$
$$- \sum_{i=1}^{n}\frac{1}{2}\varpi_i\left\|\tilde{R}_i(t)\right\|^2 + \bar{\Lambda}_i, \quad (40)$$

where $\kappa = \begin{bmatrix} \bar{L}_q^2 & \bar{L}_q^2 \\ \bar{L}_q^2 & \bar{L}_q^2 \end{bmatrix}$, $\Omega = \begin{bmatrix} 0_n & \bar{L}_q \\ \bar{L}_q & \bar{L}_q \end{bmatrix}$. Let $\xi_i > \frac{\lambda_{\max}^N + \frac{\varsigma}{2}\lambda_{\max}^M}{\lambda_{\min}^\kappa}$, where $\lambda_{\min}^\kappa$ is the minimum eigenvalue of the matrix $\kappa$, $\lambda_{\max}^N$ is the maximum eigenvalue of the matrix $N$, and $\lambda_{\max}^P$ is the maximum eigenvalue of the matrix $P$. The value $\varsigma = \min\{\sigma_1 m_1, \ldots, \sigma_n m_n\}$.

$$\dot{V}(t) \le -\frac{\varsigma}{2}\tilde{Z}^T(t)[M\otimes I_m]\tilde{Z}(t) - \frac{\varsigma}{2}\sum_{i=1}^{n}\left\|\tilde{R}_i(t)\right\|^2 + \bar{\Lambda}_i$$
$$\le -\varsigma V(t) + \bar{\Lambda}_i \quad (41)$$

The derivation is obtained from this:

$$V(t) \le V(0)e^{-\varsigma t} + \frac{\bar{\Lambda}}{\varsigma}\left(1 - e^{-\varsigma t}\right). \quad (42)$$

Eventually, the multi-USV systems is stabilized through a series of parameter adjustments. Meanwhile, it shows that the setup of the formation, collision and obstacle avoidance controller of the multi-USV systems under GPS attacks can make the formation converge to the desired position.

## IV. SIMULATION

In this section, the effectiveness of the proposed formation control strategy is verified through a numerical simulation example. There are four USVs in the simulation, including one leader USV and three follower USVs, as shown in Fig.3. The model of the nonlinear dynamics for multi-USVs as described by equation (10). The mass of the USV is $m_{usv} = 18kg$, and the dimensions of the USVs are $2.0m$ long and $0.8m$ wide. The state of the leader USV is set to be: $x_l = [-5; 5; pi/4;]$, $v_l = [0.1; 0.1; 0;]$, and the initial state of the USVs is $x_1 = [0.5; 2.5; pi/2;]$, $x_2 = [-4; 7.5; pi/2;]$, and $x_3 = [-11; 2.5; pi/2;]$. In addition, the initial velocity is set to 0 for follow USVs.

The adjacent matrix $W$ and the Laplace matrix $L$ are shown below, matrix $B = [0, 0.9, 0]$.

$$W = \begin{bmatrix} 0 & 0 & 0.7 \\ 0.6 & 0 & 0.4 \\ 0 & 1 & 0 \end{bmatrix}, L = \begin{bmatrix} 0.7 & 0 & -0.7 \\ -0.6 & 1 & -0.4 \\ 0 & -0.8 & 0.8 \end{bmatrix}$$

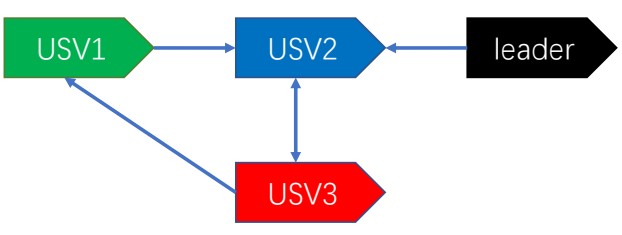

Fig. 3. Communication topology for 3 USVs and 1 leader

The mass, centrifugal and Coriolis forces, and the damping matrix for each USVs can be calculated as follows.

TABLE I
MATRIX COEFFICIENT FOR THE USV

| parameter | value | parameter | value | parameter | value |
|---|---|---|---|---|---|
| $I_z$ | 1.70 | $Y_{|z|z}$ | -2.00 | $N_{|y|z}$ | -4.00 |
| $x_g$ | 0.04 | $Y_{|y|y}$ | -36.00 | $N_{|z|z}$ | -4.00 |
| $X_x$ | -0.72 | $Y_{|y|z}$ | 2.00 | $X_{\dot{x}}$ | -2.00 |
| $X_{|x|x}$ | -1.30 | $Y_{|z|y}$ | -3.00 | $Y_{\dot{y}}$ | -10.00 |
| $X_{xxx}$ | -5.80 | $N_y$ | 0.10 | $Y_{\dot{z}}$ | 0.00 |
| $Y_y$ | -0.86 | $N_z$ | -6.00 | $N_{\dot{y}}$ | 0.00 |
| $Y_z$ | 0.10 | $N_{|y|y}$ | 5.00 | $N_{\dot{z}}$ | -1.00 |

$$M = \begin{bmatrix} 22 & 0 & 0 \\ 0 & 19 & 0.72 \\ 0 & 0.72 & 2.7 \end{bmatrix},$$

$$C = \begin{bmatrix} 0 & 0 & -19|v_{iy}| - 0.72|v_{iz}| \\ 0 & 0 & 20|v_{ix}| \\ 19|v_{iy}| + 0.72|v_{iz}| & -20|v_{ix}| & 0 \end{bmatrix},$$

$$D = \begin{bmatrix} 0.72 + 1.3\,|v_{ix}| + 5.8v_{ix}^2 & 0 \\ 0 & 0.86 + 36\,|v_{iy}| + 3\,|v_{iz}| \\ 0 & -0.1 - 5\,|v_{iy}| + 3\,|v_{iz}| \\ & 0 \\ & -0.1 - 2\,|v_{iy}| + 2\,|v_{iz}| \\ & 6 + 4\,|v_{iy}| + 4\,|v_{iz}| \end{bmatrix}.$$

The weight update law parameters are $\varpi_i = 10$, $\sigma_i = 10$. The control law parameters are set to $\Gamma = 1, i = 1,\ldots,3$. The minimum safe encounter distance and desired distance are $\rho_{\min} = 1.2$, $\rho_{des} = 1.5$. The formation pattern is described as $\Delta_1 = [4;0;0;]$, $\Delta_2 = [0;-4;0;]$, $\Delta_3 = [0;4;0;]$. Two static obstacles are centered at $[14, 37]$, $[20, 24]$. The gain coefficients $\eta = 1$, $K_{rep} = 10$, $K_r = 0.6$. The continuous function $Q(t) = \frac{t - t_s}{10}$, $t_s$ denotes the time of GPS attacks. The position Offset $\gamma = [0.05;0.05;0;]$, the positional correction vector $\tilde{\gamma} = [0.014; -0.022; 0;]$. The position threshold $\varphi = 0.02$. The simulation time is set to $[0, 49s]$.

The control inputs are bounded by:

$$\begin{cases} -350N \le u_i(1) \le 350N \\ -350N \le u_i(2) \le 350N \\ -200N \le u_i(3) \le 150N. \end{cases}$$

Fig.4 shows the trajectories of one leader USV and three USVs whether or not they were affected by GPS attacks. The dotted line indicates the formation trajectory before without GPS attacks and the solid line indicates the formation trajectory after GPS attacks. The multi-USV systems is affected by GPS attacks, the leader USV position state error exceeds a certain threshold $\varphi$ of 0.02 at 15 and 21s, the leader USV position is drastically shifted, and the formation is shifted. At 31s it starts to recover the leader trajectory and finally reaches the same target point as the formation trajectory before GPS attacks. The follow multi-USV systems can effectively track the target trajectory and maintain a great formation.

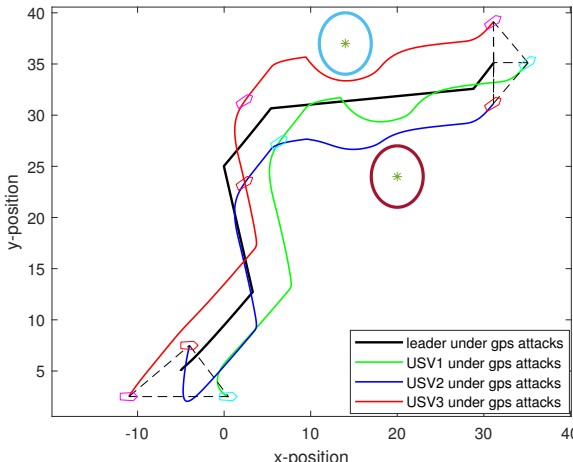

Fig. 5. The multi-USV systems formation control for collision avoidance under GPS attacks

Fig.5 shows the formation control strategy considering obstacle and collision avoidance for multi-USV systems under GPS attack. The position and velocity tracking error of the multi-USV systems in the presence of obstacles are shown in Fig.6 and Fig.7. Eventually, all the tracking errors converge to a close vicinity of 0, which means that all USVs remain at the desired position. As seen from Fig.8, given the relative distances between the multi-USVs exceed the minimum safe distance, collision between the USVs will not occur. In the illustration, $\|d_{12}\|$, $\|d_{13}\|$, and $\|d_{23}\|$ denote the relative distances between USV1 and USV2, USV1 and USV3, and USV2 and USV3, respectively.

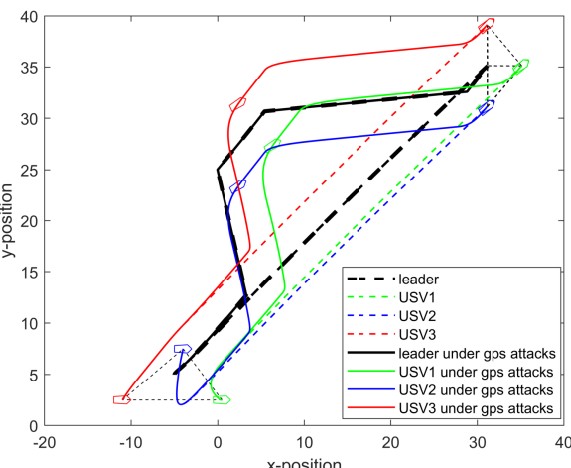

Fig. 4. The multi-USV systems formation trajectory before and after GPS attacks

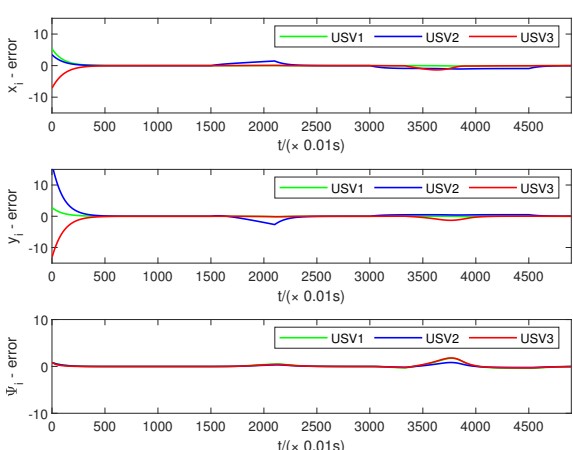

Fig. 6. The position tracking errors of multi-USVs under GPS attacks

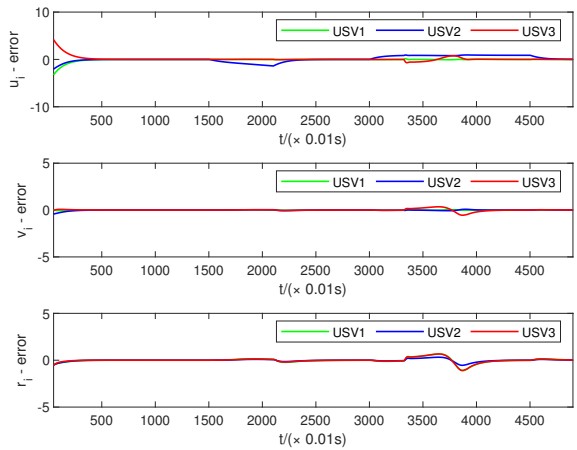

Fig. 7.  The speed tracking errors of multi-USVs under GPS attacks

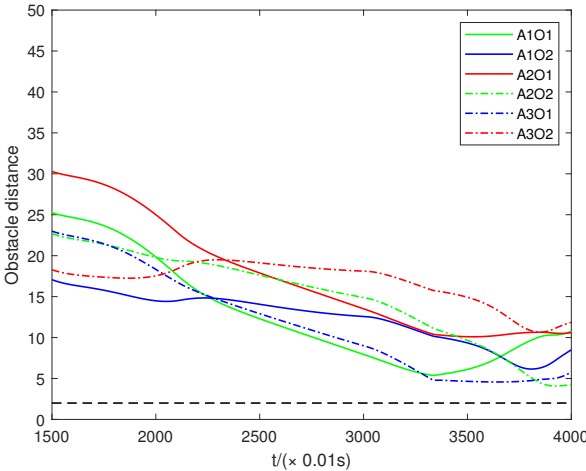

Fig. 9.  The distance between USVs and obstacles

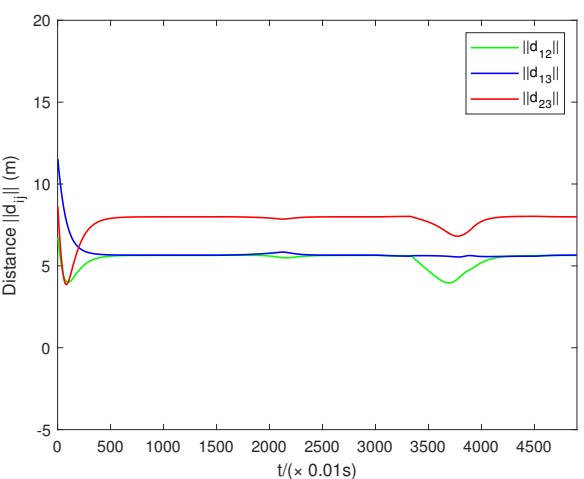

Fig. 8.  The distance between two USVs

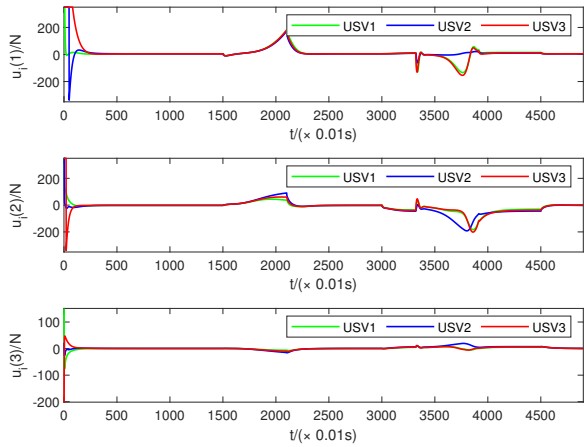

Fig. 10.  The input saturation of the multi-USV systems

As can be seen in Fig.9, given the distance between the USVs and obstacles is greater than its radius, it is demonstrated that there is no collision. The legend in Fig.9 illustrates the distance between USVs and obstacles. Fig.10 shows the dynamics of control inputs in multi-USV systems subjected to input saturation. The control inputs fluctuated slightly as a result of GPS attacks and the avoidance of obstacles. When the formation returned to normal, the system gradually stabilized.

## V.  CONCLUSION

A formation control method for USV systems under GPS spoofing attacks is proposed. In addtion, an improved APF method is utilized to guarantee that the multi-USV systems avoid obstacles and keep a collision-free distance. Finally, simulation examples demonstrate the effectiveness of the proposed formation control method.

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
