# OpenReview forum: "Improved artificial potential-based formation control of Multi-USV systems for collision and obstacle avoidance under GPS attacks"
_IEEE.org/ICIST/2024/Conference — IEEE ICIST 2024 Conference Submission_

### Official Review · Reviewer_F58B · 2024-08-21
**Accept**

**Rating:** 7
**Confidence:** 5

**Review:**

This paper presents an interesting formation control method for multi-USV (Unmanned Surface Vehicles) systems with emphasis on collision and obstacle avoidance under GPS attacks. The authors have addressed several critical challenges in a comprehensive manner, which is commendable. Here are some suggestions:1.There are many spelling mistakes in the article. Please check it carefully. For example, in the contribution part,"effectness","thoery",etc. 2.The abstract and innovation points of this paper are too short, and it is suggested to enrich the content, which fully reflects the advantages of this paper compared with the existing methods.

---

### Official Review · Reviewer_gRFE · 2024-08-23
**Accept**

**Rating:** 7
**Confidence:** 5

**Review:**

This paper uses  fuzzy logic systems to handle the external environmental disturbance and nonlinear dynamics of the multi-USV systems and designs artificial potential field (APF) method to achieve collision and obstacle  avoidance. Meanwhile,  the impact of GPS spoofing attacks on the formation control is considered. However, there are still some suggestions:
1. The innovation of the paper could be clearly expressed.
2. There are formatting issues in this paper, such as the last two paragraphs of section I.

---

### Official Review · Reviewer_4n34 · 2024-08-23
**This  is a interesting paper**

**Rating:** 8
**Confidence:** 4

**Review:**

This paper proposes a formation control method for multi-USV systems with collision and obstacle avoidance under GPS attacks. First, the external environmental disturbance and nonlinear dynamics of the multi-USV systems are handled using fuzzy logic systems. Second, a protocol for the formation control of multi-USV systems under GPS attacks is designed. The collision and obstacle avoidance issues are addressed by an improved artificial potential field (APF) method. Finally, simulation examples demonstrate the effectiveness of the proposed control method.
1). In abstract, about the proposed method, the statement is unclear. Authors need to rewrite abstract and to focus on the proposed method and to stress both the specific application and the generic aspects of the work.
2). The expression of many formulas, such as formulas (32) and (33), exceed the page margin and need to be rephrased.

---

### Decision · Program_Chairs · 2024-09-08

Accept (Oral)